# On the Performance of Battery-Assisted PS-SWIPT Enabled DF Relaying

**Zhipeng Liu** **, Guangyue Lu, Yinghui Ye ***** and Liqin Shi**

Shaanxi Key Laboratory of Information Communication Network and Security, Xi'an University of Posts and Telecommunications, Xi'an 710121, China; zhipeng_liu_steve@163.com (Z.L.); gylu@xupt.edu.cn (G.L.); liqinshi27@gmail.com (L.S.)
* Correspondence: connectyyh@126.com

**Abstract:** Compared with the conventional simultaneous wireless information and power transfer (SWIPT) based relaying with "harvest-then-forward" protocol, the battery-assisted SWIPT relaying is more practical and powerful due to the joint use of the harvested energy and supplementary battery. However, to the best of our knowledge, the performance of a battery-assisted power splitting (PS)-SWIPT decode-and-forward (DF) relay system has not been studied. In this paper, for a given amount of energy from the relay's battery, we propose to maximize the outage and ergodic capacities by optimizing the static and dynamic PS ratios that rely on statistical and instantaneous channel state information (CSI), respectively, and derive their corresponding outage and ergodic capacities. Computer simulations validate our analytical results and demonstrate the advantages of the dynamic PS over the static PS in terms of the outage and ergodic capacities, as well as the energy efficiency.

**Keywords:** decode-and-forward relay; SWIPT; power splitting; energy harvesting; battery; outage; ergodic capacity

---

## 1. Introduction

Wireless relaying is considered as a privileged means to enhance the spectral efficiency and extend the coverage of communication networks [1,2]. However, the development of relay networks is facing challenges, particularly the limited battery capacity of relay nodes. Recently, the energy harvesting (EH) technique has been incorporated into wireless relaying to prolong the operation time of the energy-constrained relay node [3]. While harvesting energy from solar, vibration or other physical phenomena is recognized as a practical solution, it may not provide an incessant and stable energy supply due to the randomness of nature resources [4]. Apart from the traditional EH approach, a new promising solution, i.e., simultaneous wireless information and power transfer (SWIPT), is proposed to harvest energy from an ambient radio-frequency (RF) signal via a power splitting (PS) or time switching (TS) scheme [5]. For the SWIPT based relay system, most existing works were based on the "harvest-then-forward" protocol, i.e., the transmit power of a relay only relies on the harvested energy, and various TS/PS schemes were proposed to improve the outage/ergodic capacity the performance (see [6–13] and reference therein).

However, due to the potentially severe fading of the wireless channel and the low efficiency of the energy harvester, the harvested energy at the relay may be insufficient to support its transmission. In order to address this issue, the authors of [14] proposed the "accumulated-then-forward" scheme for a decode-and-forward (DF) relay system, in which a rechargeable battery is deployed at the relay to assist the information transmission. In [15], the authors considered energy accumulation at the relay's battery and proposed a hybrid protocol. Recently, the authors in [16–19] proposed a battery-assisted SWIPT relay system, where the relay not only can use up the harvested energy, but also may draw

energy from its battery. Compared with the "accumulated-then-forward" scheme, the battery-assisted SWIPT relaying is easier to implement since it does not require energy accumulation.

In [17], the authors compared the outage performance between the PS scheme and the TS scheme for a battery-assisted SWIPT amplify-and-forward (AF) relay system. Recently, this work was extended to battery-assisted TS-SWIPT DF relays [18,19]. While the recent works have shown that the joint usage of the harvested energy along with the battery energy can greatly enhance the outage performance, there are still open challenges that need to be tackled. Firstly, the PS scheme for battery-assisted SWIPT DF relay systems has not been investigated yet and there is a lack of fundamental understanding of their performance, e.g., the outage and ergodic capacities. Secondly, we note that the existing works [17–19] focused on the static PS/TS scheme, where the PS/TS ratio is determined by the statistical channel state information (CSI). While it has been shown that the system performance can be further enhanced by using a dynamic PS ratio that can be adjusted based on the instantaneous CSI instead of the static one [8,11], the performance gain in battery-assisted SWIPT relay systems is not known yet. This motivates us to study the performance of dynamic PS scheme in battery-assisted SWIPT DF relay systems and compare its performance with that of a static PS scheme.

In this work, we study the performance of a battery-assisted SWIPT DF relaying, where both the static and dynamic PS ratios are considered. Our main contributions are summarized as follows.

- The static PS scheme for the battery-assisted SWIPT DF relaying is studied. In particular, for a given static PS ratio, we derive the expressions of the outage and ergodic capacities based on the statistical CSI. Using the derived results, we can determine the optimal static PS ratios that maximize the outage and ergodic capacities, respectively.
- We develop a dynamic PS scheme for the battery-assisted SWIPT DF relaying. We first derive the optimal dynamic PS ratio to maximize the outage and ergodic capacities simultaneously at each transmission slot. Using the optimal dynamic PS ratio, the expressions for the maximum outage and ergodic capacities are obtained.
- Simulation results are provided to verify our analytical results, and to compare the static and dynamic PS schemes from the following perspectives. One is to study the performance gaps (in terms of the outage and ergodic capacities) between the static and the dynamic PS schemes for a given amount of assisted energy from the relay's battery, and the other is to see how much battery energy consumption the dynamic PS scheme could save while maintaining the same performance as the static PS scheme. In addition, we compare the achievable energy efficiency between the static and dynamic PS schemes and show how the assisted energy $E_b$ affects on the energy efficiency.

## 2. System Model

As illustrated in Figure 1, we consider a dual-hop relay system, where a source $S$ with $N_s > 1$ antennas transmits its information to a single antenna destination $D$ via a single-antenna battery-assisted SWIPT DF relay. There is no direct link between the source and the destination. In our considered system, the relay not only runs out of the harvested energy, but also may extract the energy $E_b$ $(E_b \geq 0)$ from its battery. Let us focus on the worst case scenario where Rayleigh fading is used to model small-scale fading over each channel [8,10–12,16–19], and we assume that the small-scale fading follows a complex Gaussian distribution with zero mean and unit variance. Hence, the $k$-th $(k = 1, 2, \cdots, N_s)$ element $h_{1k}$ of the $S - R$ channel fading vector $\mathbf{h_1}$ is distributed as $h_{1k} \sim \mathcal{CN}\left(0, d_1^{-\alpha}\right)$, where $d_1$ is the distance of $S - R$ channel; $\alpha$ is the path loss exponent; $\mathcal{CN}\left(\cdot\right)$ denotes the complex Gaussian distribution. Likewise, the fading coefficient of $R - D$ channel follows $h_2 \sim \mathcal{CN}\left(0, d_2^{-\alpha}\right)$, where $d_2$ is the distance of $R - D$ channel.

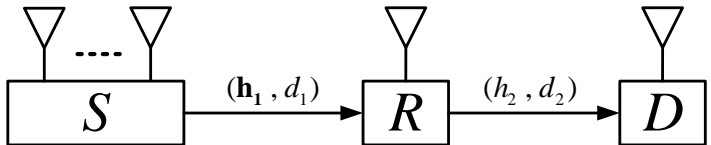

**Figure 1.** System model of dual-hop relay system.

The relay follows the PS scheme in which each transmission slot is divided into two equal sub-slots. At the first sub-slot, the source $S$ uses maximum ratio transmission (MRT) to transmit unit-energy signal $x_s$ to the relay $R$. The received signal at $R$ from $S$ is written as

$$y_{s,r} = \sqrt{P_s}\mathbf{w}^\dagger\mathbf{h_1}x_s + n_{s,r}, \tag{1}$$

where $P_s$ is the transmit power of the source $S$, $\mathbf{w}^\dagger = \mathbf{h_1^\dagger}/||\mathbf{h_1}||$ denotes the beamformer weights with conjugation operation $(\cdot)^\dagger$ and $l_2$−norm operation $||\cdot||$, and $n_{s,r}$ denotes the additive white Gaussian noise with power $\sigma^2$. Meanwhile, the relay $R$ divides the received signal into two parts through a PS ratio $\beta$: $\sqrt{\beta}y_{s,r}$ for energy harvesting and $\sqrt{1-\beta}y_{s,r}$ for information processing. Thus the harvested energy $E_h$ is written as $E_h = T\eta\beta P_s||\mathbf{w}^\dagger\mathbf{h_1}||^2/2$, where $\eta \in (0,1)$ denotes the energy conversion efficiency and the normalized $T$ (i.e., $T = 1$) is the duration time of each transmission slot. The signal-to-noise ratio (SNR) at the relay $R$, $\gamma_r$, is expressed as

$$\gamma_r = (1-\beta)\rho_s||\mathbf{w}^\dagger\mathbf{h_1}||^2, \tag{2}$$

where $\rho_s = \frac{P_s}{\sigma^2}$ denotes the input SNR.

At the second sub-slot, if the signal $x_s$ is successfully decoded, it will be forwarded to the destination $D$ by the relay $R$. The relay extracts energy $E_b$ from the battery in each transmission slot to boost its transit power, as shown in Figure 2. Accordingly, the transit power at the relay is $P_r = 2(E_h + E_b)$, and the SNR to decode $x_s$ at the destination $D$, $\gamma_d$, is calculated as

$$\gamma_d = \left(\eta\beta\rho_s||\mathbf{w}^\dagger\mathbf{h_1}||^2 + 2E_b/\sigma^2\right)|h_2|^2. \tag{3}$$

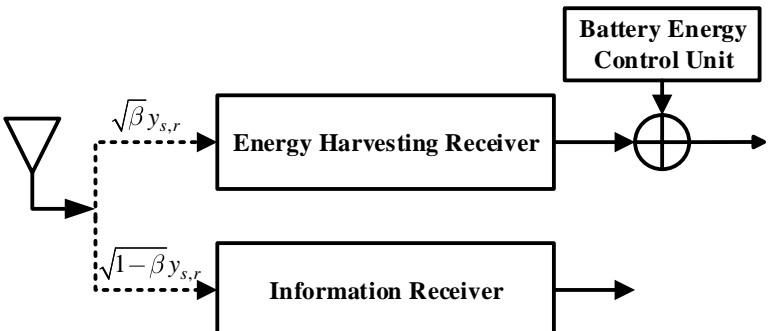

**Figure 2.** Diagram of the PS scheme at battery-assisted simultaneous wireless information and power transfer (SWIPT) enabled relay.

## 3. Static Power Splitting Scheme

### 3.1. Outage Capacity

The link is in outage when the first hop fails, or when the first hop is successful, but the second hop fails. Thus the outage probability can be written as

$$P_{out}^s = \Pr\left(\gamma_r < \gamma_{th}\right) + \Pr\left(\gamma_r > \gamma_{th}, \ \gamma_d < \gamma_{th}\right), \tag{4}$$

where $\gamma_{th} = 2^Y - 1$ denotes the SNR threshold and Y is the target rate. We define $\rho_b = \frac{2E_b}{\sigma^2}$, $X = ||\mathbf{w}^\dagger \mathbf{h_1}||^2$ and $Y = |h_2|^2$. Then the PDF of X and Y are $f_X(x) = \frac{\lambda_1^{N_s}}{\Gamma(N_s)} x^{N_s-1} e^{-\lambda_1 x}$, and $f_Y(y) = \lambda_2 e^{-\lambda_2 y}$, respectively, where $\lambda_1 = d_1^\alpha$ and $\lambda_2 = d_2^\alpha$. According to the above definitions, the outage probability is given as

$$
\begin{aligned}
P_{out}^s &= \Pr\left(X < \frac{\gamma_{th}}{(1-\beta)\rho_s}\right) \\
&\quad + \Pr\left(X > \frac{\gamma_{th}}{(1-\beta)\rho_s}, \ (\eta\beta\rho_s X + \rho_b)Y < \gamma_{th}\right) \\
&= \int_0^{\frac{\gamma_{th}}{(1-\beta)\rho_s}} f_X(x)dx + \int_{\frac{\gamma_{th}}{(1-\beta)\rho_s}}^\infty f_X(x) \int_0^{\frac{\gamma_{th}}{\eta\beta\rho_s x + \rho_b}} f_Y(y)\,dy\,dx \\
&= 1 - \frac{\lambda_1^{N_s}}{\Gamma(N_s)} \int_{\frac{\gamma_{th}}{(1-\beta)\rho_s}}^\infty x^{N_s-1} e^{-\lambda_1 x - \frac{\lambda_2 \gamma_{th}}{\eta\beta\rho_s x + \rho_b}}\,dx.
\end{aligned} \tag{5}
$$

Since the closed-form expression for (5) can not be obtained directly, the high SNR approximation derivation is as follows.

Using the Taylor series expansion for $e^{-\frac{\lambda_2 \gamma_{th}}{\eta\beta\rho_s x + \rho_b}}$, and ignoring higher order terms of $\left(\frac{\lambda_2 \gamma_{th}}{\eta\beta\rho_s x + \rho_b}\right)^2$, the outage probability at high SNR regions can be written as

$$
\begin{aligned}
P_{out}^s &\approx 1 - \frac{\lambda_1^{N_s}}{\Gamma(N_s)} \int_{\frac{\gamma_{th}}{(1-\beta)\rho_s}}^\infty x^{N_s-1} e^{-\lambda_1 x} \left(1 - \frac{\lambda_2 \gamma_{th}}{\eta\beta\rho_s x + \rho_b}\right. \\
&\quad \left. + \frac{1}{2}\left(\frac{\lambda_2 \gamma_{th}}{\eta\beta\rho_s x + \rho_b}\right)^2\right) dx \\
&= 1 - \frac{\lambda_1^{N_s}}{\Gamma(N_s)} \Bigg[ \underbrace{\int_{\frac{\gamma_{th}}{(1-\beta)\rho_s}}^\infty x^{N_s-1} e^{-\lambda_1 x} dx}_{\Xi_1}
\end{aligned} \tag{6}
$$

$$
\underbrace{- \int_{\frac{\gamma_{th}}{(1-\beta)\rho_s}}^\infty \frac{\lambda_2 \gamma_{th} x^{N_s-1} e^{-\lambda_1 x}}{\eta\beta\rho_s x + \rho_b} dx}_{\Xi_2} + \underbrace{\int_{\frac{\gamma_{th}}{(1-\beta)\rho_s}}^\infty \frac{(\lambda_2 \gamma_{th})^2 x^{N_s-1} e^{-\lambda_1 x}}{2(\eta\beta\rho_s x + \rho_b)^2} dx}_{\Xi_3} \Bigg]. \tag{7}
$$

Based on the above definitions, $\Xi_1$ can be calculated, using Equation (3.351.1) [20], as

$$\Xi_1 = \lambda_1^{-N_s} \Gamma\left(N_s, \frac{\lambda_1 \gamma_{th}}{(1-\beta)\rho_s}\right), \tag{8}$$

where $\Gamma(n,z) = \int_z^\infty u^{n-1} e^{-u} du$ is the incomplete gamma function.

The second term of $P_{out}^s$, $\Xi_2$, can be written as

$$\Xi_2 = \frac{\lambda_2 \gamma_{th}}{\eta\beta\rho_s} e^{\frac{\lambda_1 \rho_b}{\eta\beta\rho_s}} \int_a^\infty (x-b)^{N_s-1} e^{-\lambda_1 x} / x\,dx, \tag{9}$$

where $a = \frac{\gamma_{th}}{(1-\beta)\rho_s} + \frac{\rho_b}{\eta\beta\rho_s}$ and $b = \frac{\rho_b}{\eta\beta\rho_s}$. Adopting binomial expansion for $(x-b)^{Ns-1}$, $\Xi_2$ can be rewritten as

$$\Xi_2 = \frac{\lambda_2 \gamma_{th}}{\eta\beta\rho_s} e^{\frac{\lambda_1 \rho_b}{\eta\beta\rho_s}} \left[ \sum_{r=1}^{Ns-1} \binom{Ns-1}{r} (-b)^{Ns-r-1} \right.$$
$$\left. \int_a^\infty x^{r-1} e^{-\lambda_1 x} dx + (-b)^{Ns-1} \int_a^\infty e^{-\lambda_1 x}/x dx \right]. \tag{10}$$

Using Equations (3.351.2) and (3.352.2) [20], $\Xi_2$ can be calculated as

$$\Xi_2 = \frac{\lambda_2 \gamma_{th}}{\eta\beta\rho_s} e^{\frac{\lambda_1 \rho_b}{\eta\beta\rho_s}} \left[ \sum_{r=1}^{Ns-1} \binom{Ns-1}{r} (-b)^{Ns-r-1} \right.$$
$$\left. \times \lambda_1^{-r} \Gamma(r, \lambda_1 a) - (-b)^{Ns-1} \mathrm{Ei}(-\lambda_1 a) \right], \tag{11}$$

where $\mathrm{Ei}(x) = \int_{-\infty}^x \frac{1}{t} \exp(t) dt$ with $x < 0$ is the exponential integral function.

Similarly, using variable substitution and binomial expansion for $(x-b)^{Ns-1}$, the third term of $P_{out}^s$, $\Xi_3$, can be expressed as

$$\Xi_3 = \frac{1}{2} \left( \frac{\lambda_2 \gamma_{th}}{\eta\beta\rho_s} \right)^2 e^{\frac{\lambda_1 \rho_b}{\eta\beta\rho_s}} \left[ \sum_{r=2}^{Ns-1} \binom{Ns-1}{r} (-b)^{Ns-r-1} \right.$$
$$\times \int_a^\infty x^{r-2} e^{-\lambda_1 x} dx + (Ns-1)(-b)^{Ns-2}$$
$$\left. \times \int_a^\infty e^{-\lambda_1 x}/x dx + (-b)^{Ns-1} \int_a^\infty e^{-\lambda_1 x}/x^2 dx \right]. \tag{12}$$

Using Equations (3.351.2), (3.352.2) and (3.351.4) [20], $\Xi_3$ can be calculated as

$$\Xi_3 = \frac{1}{2} \left( \frac{\lambda_2 \gamma_{th}}{\eta\beta\rho_s} \right)^2 e^{\frac{\lambda_1 \rho_b}{\eta\beta\rho_s}} \left[ \sum_{r=2}^{Ns-1} \binom{Ns-1}{r} (-b)^{Ns-r-1} \right.$$
$$\times \lambda_1^{-(r-1)} \Gamma(r-1, \lambda_1 a) + \left( \lambda_1 - (Ns-1)(-b)^{Ns-2} \right)$$
$$\left. \times \mathrm{Ei}(-\lambda_1 a) + \frac{e^{-\lambda_1 a}}{a} \right]. \tag{13}$$

Substituting Systems (8), (11) and (13) into (7), the approximate expression for $P_{out}^s$ can be written as (16), as shown at the top of the next page. Accordingly, the outage capacity is calculated as

$$\tau_{out}^s = \frac{Y}{2} \left( 1 - P_{out}^s \right). \tag{14}$$

*3.2. Ergodic Capacity*

The ergodic capacity is expressed as

$$C^s = \mathrm{E} \left[ \frac{1}{2} \log_2 \left( 1 + \min(\gamma_r, \gamma_d) \right) \right] = C_1^s + C_2^s, \tag{15}$$

where $C_1^s$ and $C_2^s$ are the ergodic capacities corresponding to the case $\gamma_r < \gamma_d$ and $\gamma_r \geq \gamma_d$, respectively [10].

$C_1^s$ and $C_2^s$ can be derived as Systems (17)–(19) and Systems (20)–(22), respectively. From $\gamma_r < \gamma_d$, we can obtain that the range of random variable $Y$ is $y > \frac{(1-\beta)\rho_s x}{\eta\beta\rho_s x + \rho_b}$. Accordingly, the ergodic capacity $C_1^s$ corresponding to $\gamma_r < \gamma_d$ is presented as (17). After solving the internal integral with respect to

the variable $y$, (18) can be obtained. Since the integral is unbounded and there is no closed-form for (18), we use the variable substitution $y = \tan\theta$ and adopt the Gaussian–Chebyshev quadrature (in this paper, we adopt the Gaussian–Chebyshev quadrature instead of other approximation methods because it can provide sufficient level of accuracy with very few terms; thanks to this advantage, the Gaussian–Chebyshev quadrature has been widely used in the state-of-the-art works [11,12,21]) to approximate Systems (18) as (19), where $\omega_1 = \frac{\pi}{N}$, $f_{i_1} = \cos\left(\frac{(2i_1-1)\pi}{2N}\right)$, $\theta_{i_1} = \frac{\pi}{4}\left(f_{i_1}+1\right)$ and $\Theta_1 = \frac{\lambda_1^{N_s}\pi\omega_1}{8\Gamma(N_s)}\sqrt{1-f_{i_1}^2}$. $N$ is complexity and accuracy tradeoff parameter. Similarly, the ergodic capacity $C_2^s$ corresponding to $\gamma_r \geq \gamma_d$ is expressed as (20). Adopting the Gaussian–Chebyshev quadrature, (21) can be obtained. Similar to (19), (21) can be approximated as (22) by using the Gaussian–Chebyshev quadrature, where $\omega_2 = \omega_3 = \frac{\pi}{N}$, $f_{i_2} = \cos\left(\frac{(2i_2-1)\pi}{2N}\right)$, $f_{i_3} = \cos\left(\frac{(2i_3-1)\pi}{2N}\right)$, $c_{i_2} = f_{i_2} + 1$, $\theta_{i_3} = \frac{\pi}{4}\left(f_{i_3}+1\right)$, $\Phi_{i_3} = \frac{(1-\beta)\rho_s\tan\theta_{i_3}}{\eta\beta\rho_s\tan\theta_{i_3}+\rho_b}$, $\Theta_2 = \frac{\lambda_1^{N_s}\lambda_2\omega_2}{4\Gamma(N_s)}\sqrt{1-f_{i_2}^2}$ and $\Theta_3 = \frac{\lambda_1^{N_s}\lambda_2\pi\omega_2\omega_3}{16\Gamma(N_s)}\sqrt{1-f_{i_3}^2}\sqrt{1-f_{i_2}^2}$. Based on (15), (19) and (22), we can obtain the ergodic capacity as given in (23) at the top of the next page.

$$
\begin{aligned}
P_{out}^s \approx &\, 1 - \frac{\lambda_1^{N_s}}{\Gamma(N_s)}\left[\Xi_1 - \Xi_2 + \Xi_3\right] \\
\approx &\, 1 - \frac{\lambda_1^{N_s}}{\Gamma(N_s)}\left[\lambda_1^{-N_s}\Gamma\left(N_s, \frac{\lambda_1\gamma_{th}}{(1-\beta)\rho_s}\right)\right. \\
&- \frac{\lambda_2\gamma_{th}}{\eta\beta\rho_s}e^{\frac{\lambda_1\rho_b}{\eta\beta\rho_s}}\left(\sum_{r=1}^{Ns-1}\binom{Ns-1}{r}(-b)^{Ns-r-1}\lambda_1^{-r}\Gamma(r,\lambda_1 a) - (-b)^{Ns-1}\mathrm{Ei}(-\lambda_1 a)\right) \\
&+ \frac{1}{2}\left(\frac{\lambda_2\gamma_{th}}{\eta\beta\rho_s}\right)^2 e^{\frac{\lambda_1\rho_b}{\eta\beta\rho_s}}\left(\sum_{r=2}^{Ns-1}\binom{Ns-1}{r}(-b)^{Ns-r-1}\lambda_1^{-(r-1)}\Gamma(r-1,\lambda_1 a)\right. \\
&\left.\left.+ \left(\lambda_1 - (Ns-1)(-b)^{Ns-2}\right)\mathrm{Ei}(-\lambda_1 a) + \frac{e^{-\lambda_1 a}}{a}\right)\right]
\end{aligned}
$$
(16)

$$
C_1^s = \frac{1}{2}\int_0^\infty \int_{\frac{(1-\beta)\rho_s x}{\eta\beta\rho_s x+\rho_b}}^\infty \frac{\lambda_1^{N_s}}{\Gamma(N_s)}x^{N_s-1}e^{-\lambda_1 x}\lambda_2 e^{-\lambda_2 y}\log_2(1+(1-\beta)\rho_s x)\,dy\,dx
$$
(17)

$$
= \frac{\lambda_1^{N_s}}{2\Gamma(N_s)}\int_0^\infty x^{N_s-1}e^{-\left(\lambda_1 x + \frac{(1-\beta)\lambda_2\rho_s x}{\eta\beta\rho_s x+\rho_b}\right)}\log_2(1+(1-\beta)\rho_s x)\,dx
$$
(18)

$$
\approx \sum_{i_1=1}^N \Theta_1 \tan^{N_s-1}\theta_{i_1} e^{-\left(\lambda_1\tan\theta_{i_1} + \frac{(1-\beta)\lambda_2\rho_s\tan\theta_{i_1}}{\eta\beta\rho_s\tan\theta_{i_1}+\rho_b}\right)}\log_2(1+(1-\beta)\rho_s\tan\theta_{i_1})\sec^2\theta_{i_1}
$$
(19)

$$
C_2^s = \frac{1}{2}\int_0^\infty \int_0^{\frac{(1-\beta)\rho_s x}{\eta\beta\rho_s x+\rho_b}} \frac{\lambda_1^{N_s}}{\Gamma(N_s)}x^{N_s-1}e^{-\lambda_1 x}\lambda_2 e^{-\lambda_2 y}\log_2(1+(\eta\beta\rho_s x+\rho_b)y)\,dy\,dx
$$
(20)

$$
\approx \int_0^\infty \sum_{i_2=1}^N \Theta_2 \frac{(1-\beta)\rho_s x}{\eta\beta\rho_s x+\rho_b}x^{N_s-1}e^{-\left(\lambda_1 x + \frac{(1-\beta)\lambda_2 c_{i_2}\rho_s x}{2(\eta\beta\rho_s x+\rho_b)}\right)}\log_2\left(1+\frac{(\eta\beta\rho_s x+\rho_b)(1-\beta)c_{i_2}\rho_s x}{2(\eta\beta\rho_s x+\rho_b)}\right)dx
$$
(21)

$$
\approx \sum_{i_3=1}^N \sum_{i_2=1}^N \Theta_3 \Phi_{i_3}\tan^{N_s-1}\theta_{i_3}\sec^2\theta_{i_3}e^{-\left(\lambda_1\tan\theta_{i_3} + \frac{\lambda_2 c_{i_2}\Phi_{i_3}}{2}\right)}\log_2\left(1+\frac{c_{i_2}\Phi_{i_3}(\eta\beta\rho_s\tan\theta_{i_3}+\rho_b)}{2}\right)
$$
(22)

**Remark 1.** *The derived expressions Equations (14) and (23) can serve the following purposes. Firstly, the derived results can be used to obtain accurate outage and ergodic capacities instead of the computer simulations. Secondly, since the optimal static PS ratio is determined by the statistic channel gains instead of instantaneous channel gains, it is practical to obtain the optimal static ratio offline by using the derived expressions and such an approach has been widely adopted in many works [10–12]. Lastly, we can obtain some insights such as how $E_b$ affects the optimal PS ratio in terms of energy efficiency, and how to select the assisted energy $E_b$ to realize a higher energy efficiency.*

$$C^s \approx \sum_{i_1=1}^{N} \Theta_1 \tan^{N_s-1}\theta_{i_1} e^{-\left(\lambda_1 \tan\theta_{i_1} + \frac{(1-\beta)\lambda_2\rho_s \tan\theta_{i_1}}{\eta\beta\rho_s \tan\theta_{i_1} + \rho_b}\right)} \log_2\left(1 + (1-\beta)\rho_s \tan\theta_{i_1}\right) \sec^2\theta_{i_1}$$

$$+ \sum_{i_3=1}^{N} \sum_{i_2=1}^{N} \Theta_3 \Phi_{i_3} \tan^{N_s-1}\theta_{i_3} \sec^2\theta_{i_3} e^{-\left(\lambda_1 \tan\theta_{i_3} + \frac{\lambda_2 c_{i_2}\Phi_{i_3}}{2}\right)} \log_2\left(1 + \frac{c_{i_2}\Phi_{i_3}\left(\eta\beta\rho_s \tan\theta_{i_3} + \rho_b\right)}{2}\right) \quad (23)$$

## 4. Dynamic Power Splitting Scheme

Since maximizing the instantaneous capacity is able to maximize the outage and ergodic capacities simultaneously [8], the optimal dynamic PS ratio at each transmission slot can be obtained by solving the following problem,

$$C = \max_{0 \le \beta \le 1} \min\left\{\frac{1}{2}\log_2\left(1+\gamma_r\right), \frac{1}{2}\log_2\left(1+\gamma_d\right)\right\}. \quad (24)$$

The capacity optimization problem is equivalent to the following problem: $\gamma^{op} = \max_{0 \le \beta \le 1} \min\left(\gamma_r, \gamma_d\right)$.

**Lemma 1.** *The optimal dynamic PS ratio $\beta^*$ is*

$$\beta^* = \begin{cases} 0 & , \quad \rho_s||\mathbf{w}^\dagger\mathbf{h_1}||^2 < \rho_b|h_2|^2, \\ \frac{\rho_s||\mathbf{w}^\dagger\mathbf{h_1}||^2 - \rho_b|h_2|^2}{\rho_s||\mathbf{w}^\dagger\mathbf{h_1}||^2(\eta|h_2|^2+1)}, & \rho_s||\mathbf{w}^\dagger\mathbf{h_1}||^2 \ge \rho_b|h_2|^2. \end{cases} \quad (25)$$

**Proof.** It is clear that $\gamma_r$ and $\gamma_d$ are the monotone decreasing and increasing functions with respect to $\beta$, respectively. When $\rho_s||\mathbf{w}^\dagger\mathbf{h_1}||^2 < \rho_b|h_2|^2$, curves $\gamma_r$ and $\gamma_d$ have no intersection, and the optimal SNR $\gamma_1^{op}$ is equal to $\rho_s||\mathbf{w}^\dagger\mathbf{h_1}||^2$ when $\beta^* = 0$. When $\rho_s||\mathbf{w}^\dagger\mathbf{h_1}||^2 \ge \rho_b|h_2|^2$, the optimal SNR is achievable when $\gamma_r$ equals $\gamma_d$. In this case, $0 \le \beta^* = \frac{\rho_s||\mathbf{w}^\dagger\mathbf{h_1}||^2 - \rho_b|h_2|^2}{\rho_s||\mathbf{w}^\dagger\mathbf{h_1}||^2(\eta|h_2|^2+1)} < 1$. Substituting $\beta^*$ into (2), the corresponding optimal SNR can be given as $\gamma_2^{op} = \frac{\left(\rho_b + \eta\rho_s||\mathbf{w}^\dagger\mathbf{h_1}||^2\right)|h_2|^2}{1+\eta|h_2|^2}$. $\square$

**Remark 2.** *As exhibited in (25), when $\rho_s||\mathbf{w}^\dagger\mathbf{h_1}||^2 < \rho_b|h_2|^2$ is satisfied, all the received power at the relay is used for information decoding and the transit power at the relay only comes from the assisted battery. In this case, the relay system degenerates into a traditional relay system without SWIPT. When $\rho_s||\mathbf{w}^\dagger\mathbf{h_1}||^2 \ge \rho_b|h_2|^2$ holds, the better the $S-R$ ($R-D$) channel condition is, the larger the proportion of the received power at the relay forwarded to the energy harvesting (information decoding) circuit is. Moreover, we note that our proposed dynamic PS can be reduced to the one in [8] when $E_b = 0$ and $N_s = 1$, that is to say, our study is more generally than [8]. Therefore, by adjusting system parameters, the performance analysis of the proposed scheme is also applicable to that of [8].*

### 4.1. Outage Capacity for the Optimal Dynamic PS

The outage probability for the dynamic PS is given as $P_{out} = \Pr(\min\left(\gamma_r, \gamma_d\right) < \gamma_{th})$. Thus, the outage probability corresponding to the optimal dynamic PS can be written as

$$P_{out}^d = \Pr\left(\gamma^{op} < \gamma_{th}\right) = \Pr\left(\gamma_1^{op} < \gamma_{th}, \beta^* = 0\right)$$

$$+ \Pr\left(\gamma_2^{op} < \gamma_{th}, \beta^* = \frac{\rho_s||\mathbf{w}^\dagger\mathbf{h_1}||^2 - \rho_b|h_2|^2}{\rho_s||\mathbf{w}^\dagger\mathbf{h_1}||^2\left(\eta|h_2|^2+1\right)}\right). \quad (26)$$

For notational simplicity, denote the first and the second terms of (26) as $P_1^d$ and $P_2^d$, respectively. Then we process (26) from whether the relay draws the energy from its battery or not.

### 4.1.1. $\rho_b = 0$

This case means that the energy $E_b$ extracted from the battery equals 0, inferring $P_1^d = 0$. Accordingly, $P_{out}^d$ can be given as

$$P_{out}^d = P_2^d = \Pr\left(x < \frac{\gamma_{th}\left(1 + \eta y\right)}{\eta \rho_s y}\right)$$

$$= \int_0^\infty \int_0^{\frac{\gamma_{th}(1+\eta y)}{\eta \rho_s y}} \frac{\lambda_1^{N_s}}{\Gamma\left(N_s\right)} x^{N_s - 1} e^{-\lambda_1 x} \lambda_2 e^{-\lambda_2 y} dx dy$$

$$= \frac{\lambda_2}{\Gamma\left(N_s\right)} \int_0^\infty \gamma\left(N_s, \frac{\lambda_1 \gamma_{th}\left(1 + \eta y\right)}{\eta \rho_s y}\right) e^{-\lambda_2 y} dy, \tag{27}$$

where $\gamma\left(n, z\right) = \int_0^z u^{n-1} e^{-u} du$ is the incomplete gamma function. $P_{out}^d$ at high SNR regions can be given by

$$P_{out}^d \approx \frac{\lambda_2}{\Gamma\left(N_s\right)} \int_0^\infty \left(\Gamma\left(N_s\right) - \Gamma\left(N_s, \frac{\lambda_1 \gamma_{th}}{\eta \rho_s y}\right)\right) e^{-\lambda_2 y} dy \tag{28}$$

$$= 1 - \frac{2(\lambda_1 \lambda_2 \gamma_{th} / \left(\eta \rho_s\right))^{\frac{N_s}{2}}}{\Gamma\left(N_s\right)} K_{N_s}\left(2\sqrt{\frac{\lambda_1 \lambda_2 \gamma_{th}}{\eta \rho_s}}\right), \tag{29}$$

where $K_n\left(v\right)$ denotes modified Bessel function of the second kind. Using Equation (6.453) [20], (29) can be obtained.

### 4.1.2. $\rho_b \neq 0$

In this case, the corresponding $P_{out}^d$ can be calculated as

$$P_{out}^d = P_1^d + P_2^d = \Pr\left(x < \frac{\gamma_{th}}{\rho_s}, y > \frac{\rho_s x}{\rho_b}\right)$$

$$+ \Pr\left(x < \frac{\gamma_{th} + \left(\eta \gamma_{th} - \rho_b\right) y}{\eta \rho_s y}, x > \frac{\rho_b y}{\rho_s}\right). \tag{30}$$

The first term $P_1^d$ can be calculated, using Equation (3.351.1) [20], as

$$P_1^d = \int_0^{\frac{\gamma_{th}}{\rho_s}} \int_{\frac{\rho_s x}{\rho_b}}^\infty \frac{\lambda_1^{N_s}}{\Gamma\left(N_s\right)} x^{N_s - 1} e^{-\lambda_1 x} \lambda_2 e^{-\lambda_2 y} dy dx$$

$$= \frac{\lambda_1^{N_s}}{\Gamma\left(N_s\right)} \int_0^{\frac{\gamma_{th}}{\rho_s}} x^{N_s - 1} e^{-\left(\lambda_1 + \frac{\lambda_2 \rho_s}{\rho_b}\right) x} dx$$

$$= \left(\frac{\lambda_1 \rho_b}{\psi_1}\right)^{N_s} \frac{\gamma\left(N_s, \gamma_{th} \psi_1 / \left(\rho_b \rho_s\right)\right)}{\Gamma\left(N_s\right)}, \tag{31}$$

where $\psi_1 = \lambda_1 \rho_b + \lambda_2 \rho_s$. The second term $P_2^d$ can be calculated as

$$P_2^d = \Pr\left(\psi_2 < x < \psi_3, \psi_3 > \psi_2\right), \tag{32}$$

where $\psi_2 = \frac{\rho_b y}{\rho_s}$ and $\psi_3 = \frac{\gamma_{th} + \left(\eta \gamma_{th} - \rho_b\right) y}{\eta \rho_s y}$. When $\psi_3 > \psi_2$, we obtain the following inequality, given by

$$\eta \rho_b y^2 + \left(\rho_b - \eta \gamma_{th}\right) y - \gamma_{th} < 0. \tag{33}$$

Combining (33) and $y > 0$, the range of $y$ is $0 < y < \frac{\gamma_{th}}{\rho_b}$. Thus, $P_2^d$ can be recalculated, using Equation (3.382.5) [20], as

$$
\begin{aligned}
P_2^d &= \int_0^{\frac{\gamma_{th}}{\rho_b}} \int_{\psi_2}^{\psi_3} \frac{\lambda_1^{N_s}}{\Gamma(N_s)} x^{N_s-1} e^{-\lambda_1 x} \lambda_2 e^{-\lambda_2 y} dx dy \\
&= \frac{\lambda_2 e^{-\lambda_1 \psi_2}}{\Gamma(N_s)} \int_0^{\frac{\gamma_{th}}{\rho_b}} e^{-\lambda_2 y} \int_0^{\lambda_1(\psi_3-\psi_2)} (x + \lambda_1\psi_2)^{N_s-1} e^{-x} dx dy \\
&= \frac{\lambda_2}{\Gamma(N_s)} \int_0^{\frac{\gamma_{th}}{\rho_b}} (\gamma(N_s, \lambda_1\psi_3) - \gamma(N_s, \lambda_1\psi_2)) e^{-\lambda_2 y} dy.
\end{aligned}
\tag{34}
$$

Using the Gaussian–Chebyshev quadrature, the $P_2^d$ can be the approximated as

$$
\begin{aligned}
P_2^d \approx \sum_{i_4=1}^{N} \Theta_4 e^{-\lambda_2 \Phi_{i_4}} \Bigg[ &\gamma\left(N_s, \frac{\lambda_1(\Phi_{i_4}(\eta\gamma_{th} - \rho_b) + \gamma_{th})}{\eta\rho_s\Phi_{i_4}}\right) \\
&- \gamma\left(N_s, \frac{\lambda_1\rho_b\Phi_{i_4}}{\rho_s}\right)\Bigg],
\end{aligned}
\tag{35}
$$

where $\omega_4 = \frac{\pi}{N}$, $f_{i_4} = \cos\left(\frac{(2i_4-1)\pi}{2N}\right)$, $\Phi_{i_4} = \frac{\gamma_{th}}{2\rho_b}(f_{i_4}+1)$, $\Theta_4 = \frac{\lambda_2\gamma_{th}\omega_4}{2\rho_b\Gamma(N_s)}\sqrt{1-f_{i_4}^2}$.

Based on the above discussions, the minimum outage probability $P_{out}^d$ can be expressed as

$$
P_{out}^d = \begin{cases}
1 - \frac{2(\lambda_1\lambda_2\gamma_{th}/(\eta\rho_s))^{\frac{N_s}{2}}}{\Gamma(N_s)} K_{N_s}\left(2\sqrt{\frac{\lambda_1\lambda_2\gamma_{th}}{\eta\rho_s}}\right) & , \rho_b = 0, \\[2ex]
\left(\frac{\lambda_1\rho_b}{\psi_1}\right)^{N_s} \frac{\gamma(N_s,\gamma_{th}\psi_1/(\rho_b\rho_s))}{\Gamma(N_s)} + \sum_{i_4=1}^{N} \Theta_4 e^{-\lambda_2\Phi_{i_4}} \\
\times \left[\gamma\left(N_s, \frac{\lambda_1(\Phi_{i_4}(\eta\gamma_{th} - \rho_b) + \gamma_{th})}{\eta\rho_s\Phi_{i_4}}\right) - \gamma\left(N_s, \frac{\lambda_1\rho_b\Phi_{i_4}}{\rho_s}\right)\right] & , \rho_b \neq 0.
\end{cases}
\tag{36}
$$

Accordingly, the outage capacity for a given assisted energy $E_b$ corresponding to the optimal dynamic PS is calculated as

$$
\tau_{out}^d = \frac{\mathrm{Y}}{2}\left(1 - P_{out}^d\right).
\tag{37}
$$

*4.2. Ergodic Capacity for the Optimal Dynamic PS*

According to (15), the ergodic capacity for a fixed assisted energy $E_b$ corresponding to the optimal dynamic PS is

$$
C^d = \mathrm{E}\left[\frac{1}{2}\log_2(1 + \gamma^{op})\right] = C_1^d + C_2^d,
\tag{38}
$$

where $C_1^d$ and $C_2^d$ are the ergodic capacities when $\beta^* = 0$ and $\beta^* = \frac{\rho_s||\mathbf{w}^\dagger\mathbf{h_1}||^2 - \rho_b|h_2|^2}{\rho_s||\mathbf{w}^\dagger\mathbf{h_1}||^2(\eta|h_2|^2+1)}$, respectively.

$C_1^d$ and $C_2^d$ can be derived as $(39)-(41)$ and $(42)-(44)$, respectively. According to $\beta^* = 0$, we can get that the range of random variable $X$ is $x < \frac{\rho_b y}{\rho_s}$. Then the ergodic capacity $C_1^d$ corresponding to $\beta^* = 0$ is presented as (39). Adopting the Gaussian–Chebyshev quadrature, (40) can be obtained. Similar to (21), (40) can be approximated as (41), where $\omega_5 = \omega_6 = \frac{\pi}{N}$, $f_{i_5} = \cos\left(\frac{(2i_5-1)\pi}{2N}\right)$, $f_{i_6} = \cos\left(\frac{(2i_6-1)\pi}{2N}\right)$, $c_{i_5} = f_{i_5} + 1$, $\theta_{i_6} = \frac{\pi}{4}(f_{i_6}+1)$ and $\Theta_5 = \frac{(\lambda_1\rho_b/2\rho_s)^{N_s}\lambda_2\pi\omega_5\omega_6}{8\Gamma(N_s)}\sqrt{1-f_{i_6}^2}\sqrt{1-f_{i_5}^2}$. Similarly, the ergodic capacity $C_2^d$ corresponding to $\beta^* = \frac{\rho_s||\mathbf{w}^\dagger\mathbf{h_1}||^2 - \rho_b|h_2|^2}{\rho_s||\mathbf{w}^\dagger\mathbf{h_1}||^2(\eta|h_2|^2+1)}$ is expressed as (42). By changing the variable, we can obtain (43). Similar to (40), (43) can be approximated as (44) by using the Gaussian–Chebyshev quadrature again, where $\omega_7 = \omega_8 = \frac{\pi}{N}$, $f_{i_7} = \cos\left(\frac{(2i_7-1)\pi}{2N}\right)$, $f_{i_8} = \cos\left(\frac{(2i_8-1)\pi}{2N}\right)$, $\theta_{i_7} = \frac{\pi}{4}(f_{i_7}+1)$,

$\theta_{i_8} = \frac{\pi}{4}\left(f_{i_8}+1\right)$, $\Phi_{i_{7,8}} = \tan\theta_{i_7} + \rho_b\tan\theta_{i_8}/\rho_s$ and $\Theta_6 = \frac{\lambda_1^{N_s}\lambda_2\pi^2\omega_7\omega_8}{32\Gamma(N_s)}\sqrt{1-f_{i_8}^2}\sqrt{1-f_{i_7}^2}$. Based on (38), (41) and (44), we can obtain the ergodic capacity as shown in (45).

$$C_1^d = \frac{1}{2}\int_0^\infty \int_0^{\frac{\rho_b y}{\rho_s}} \frac{\lambda_1^{N_s}}{\Gamma(N_s)} x^{N_s-1}e^{-\lambda_1 x}\lambda_2 e^{-\lambda_2 y}\log_2\left(1+\rho_s x\right)dxdy \tag{39}$$

$$\approx \frac{(\lambda_1\rho_b/2\rho_s)^{N_s}\lambda_2\omega_5}{2\Gamma(N_s)}\int_0^\infty \sum_{i_5=1}^N \sqrt{1-f_{i_5}^2}c_{i_5}^{N_s-1}y^{N_s}e^{-\left(\frac{\lambda_1\rho_b c_{i_5}}{2\rho_s}+\lambda_2\right)y}\log_2\left(1+\frac{\rho_b c_{i_5}y}{2}\right)dy \tag{40}$$

$$\approx \sum_{i_6=1}^N \sum_{i_5=1}^N \Theta_5\tan^{N_s}\theta_{i_6}\sec^2\theta_{i_6}c_{i_5}^{N_s-1}e^{-\left(\frac{\lambda_1\rho_b c_{i_5}}{2\rho_s}+\lambda_2\right)\tan\theta_{i_6}}\log_2\left(1+\frac{\rho_b c_{i_5}\tan\theta_{i_6}}{2}\right) \tag{41}$$

$$C_2^d = \frac{1}{2}\int_0^\infty \int_{\frac{\rho_b y}{\rho_s}}^\infty \frac{\lambda_1^{N_s}}{\Gamma(N_s)}x^{N_s-1}e^{-\lambda_1 x}\lambda_2 e^{-\lambda_2 y}\log_2\left(1+\frac{y(\rho_b+\eta\rho_s x)}{1+\eta y}\right)dxdy \tag{42}$$

$$= \frac{1}{2}\int_0^\infty \int_0^\infty \frac{\lambda_1^{N_s}}{\Gamma(N_s)}\left(x+\frac{\rho_b y}{\rho_s}\right)^{N_s-1}e^{-\lambda_1\left(x+\frac{\rho_b y}{\rho_s}\right)}\lambda_2 e^{-\lambda_2 y}\log_2\left(1+\left(\rho_b+\eta\rho_s\left(x+\frac{\rho_b y}{\rho_s}\right)\right)\right.$$

$$\left. \times y/(1+\eta y)\right)dxdy \tag{43}$$

$$\approx \sum_{i_8=1}^N \sum_{i_7=1}^N \Theta_6\sec^2\theta_{i_8}\sec^2\theta_{i_7}\Phi_{i_{7,8}}^{N_s-1}e^{-\left(\lambda_1\Phi_{i_{7,8}}+\lambda_2\tan\theta_{i_8}\right)}\log_2\left(1+\frac{\tan\theta_{i_8}\left(\rho_b+\eta\rho_s\Phi_{i_{7,8}}\right)}{1+\eta\tan\theta_{i_8}}\right) \tag{44}$$

$$C^d \approx \sum_{i_6=1}^N \sum_{i_5=1}^N \Theta_5\tan^{N_s}\theta_{i_6}\sec^2\theta_{i_6}c_{i_5}^{N_s-1}e^{-\left(\frac{\lambda_1\rho_b c_{i_5}}{2\rho_s}+\lambda_2\right)\tan\theta_{i_6}}\log_2\left(1+\frac{\rho_b c_{i_5}\tan\theta_{i_6}}{2}\right)$$

$$+ \sum_{i_8=1}^N \sum_{i_7=1}^N \Theta_6\sec^2\theta_{i_8}\sec^2\theta_{i_7}\Phi_{i_{7,8}}^{N_s-1}e^{-\left(\lambda_1\Phi_{i_{7,8}}+\lambda_2\tan\theta_{i_8}\right)}\log_2\left(1+\frac{\tan\theta_{i_8}\left(\rho_b+\eta\rho_s\Phi_{i_{7,8}}\right)}{1+\eta\tan\theta_{i_8}}\right) \tag{45}$$

## 5. Simulation Results

In this section, we present simulation results to verify the above analysis. The simulation parameters are set as follows [18]: $\eta=0.5$, $d_1=2$, $d_2=2$, $\alpha=3$, $N=10$ and $N_s=3$.

Figure 3 plots the relative approximate error versus $\rho_s$ with different setting of $E_b$ to illustrate the accuracy of the Taylor series expansion approach. Specifically, according to [22], the relative approximate error can be computed as

$$\delta = \left|\frac{\text{analytical result} - \text{simulation result}}{\text{simulation result}}\right|, \tag{46}$$

where the analytical result is obtained from (14) and the simulation result is achieved by Monte-Carlo simulations. As presented in this figure, the relative approximate error approaches zero with the increase of $\rho_s$. For example, when $E_b = 0.1 \times 10^{-6}$ J and $\rho_s = 34$ dB, the relative approximation error $\delta$ is 0.00455, which provides enough accuracy for the outage capacity. Thus, our derived expressions based on the Taylor series approximation can evaluate the outage performance of the considered network effectively.

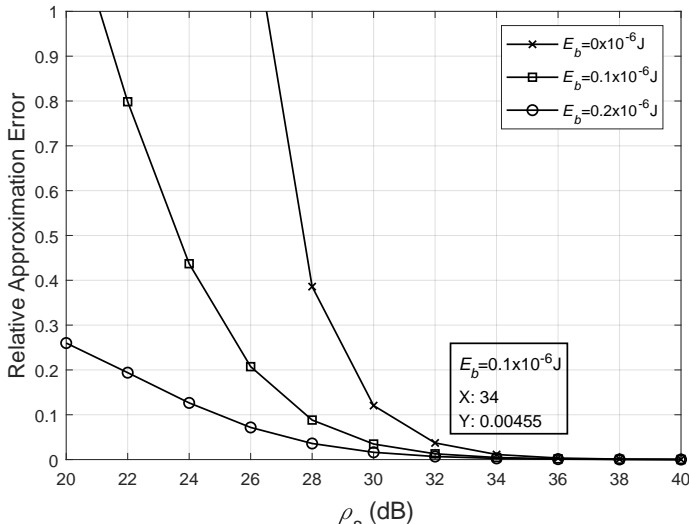

**Figure 3.** Relative approximate error vs. $\rho_s$.

Figure 4 shows the relative approximate error versus $N$ with different setting of $E_b$ to verify the accuracy of the Gaussian–Chebyshev approach. Similarly, the analytical result is obtained from (23) and the simulation result is achieved by Monte-Carlos simulations. As shown in this figure, when $N > 7$, the approximate results are sufficiently accurate. Thus, our derived expressions based on the Gaussian–Chebyshev approach can also evaluate the outage/ergodic performance of the investigated network efficiently.

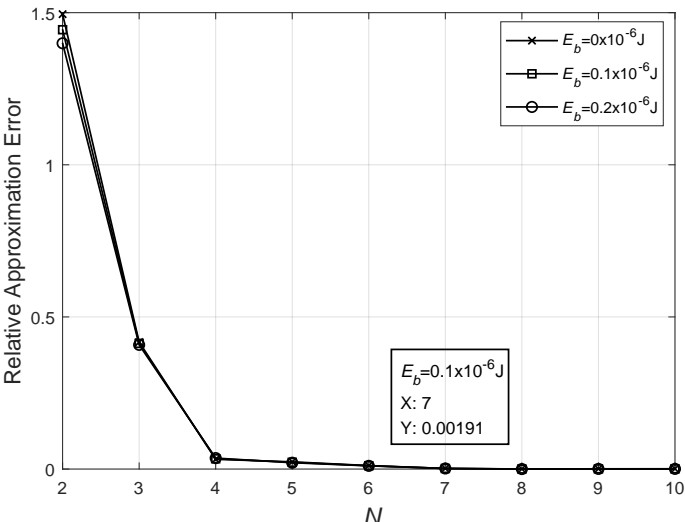

**Figure 4.** Relative approximate error vs. $N$.

Figure 5 plots the outage capacity against the static PS ratio $\beta$ with the given energy $E_b$. The approximated results match closely to simulations, which validates the accuracy of derivations in (5). Obviously, the outage capacity is a concave function with respect to $\beta$, and the optimal static PS ratio can be obtain by one-dimensional search methods. On one hand, we can see that the outage capacity for the fixed $\beta$ increases with the increase of $E_b$ due to the improvement of the transit power at the relay. On the other hand, we can also see that the optimal static PS ratio to maximize the outage capacity decreases as $E_b$ increases due to lower dependence on EH.

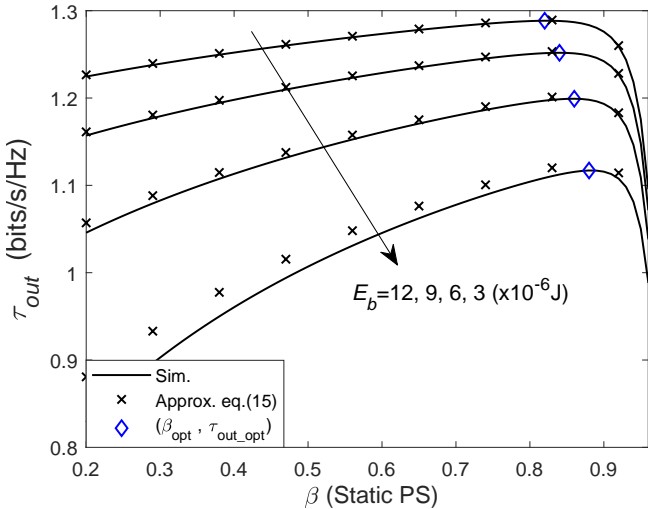

**Figure 5.** The outage capacity vs. $\beta$, where $\rho_s = 30$ dB and Y $= 3$ bits/s/Hz.

Figure 6 shows the variations of the ergodic capacity for the fixed energy $E_b$ with the static PS ratio $\beta$. From this figure, we can see that the derived approximation results under the expression in (23) match with the simulation results. It is also shown that the ergodic capacity is a concave function of $\beta$. Likewise, the optimal static PS ratio that maximizes the ergodic capacity can be determined by search techniques. It can be observed that the ergodic capacity for the fixed $\beta$ increases as $E_b$ increases. Moreover, it can also be observed that an increase in $E_b$ causes the optimal static PS ratio to decrease.

Figure 7 depicts the outage capacity against the assisted energy $E_b$ for three PS schemes. For the random PS scheme, the PS ratio is randomly given from the interval [0,1]. The approximate results match with the simulations results well, which demonstrates the correctness of our derivations. One can see that for a given $E_b$, the dynamic PS can achieve the highest outage capacity and the performance of random static PS is the worst. This is because the dynamic PS maximizes the overall SNR at each transmission slot by using instantaneous CSI, while the optimal static PS only uses the statistic CSI and no CSI is required for the random scheme. This observation also shows the importance of selecting the appropriated static PS ratio. In addition, it can also be seen that for the same outage capacity, the dynamic PS can consume much less the assisted energy from the battery compared with the optimal static one. The reason is as follows. It can be seen from (25) that the adjustment of the optimal dynamic PS ratio is based on not only the instantaneous CSI but also the assisted energy $E_b$. This allows $E_b$ to be fully utilized at each time slot.

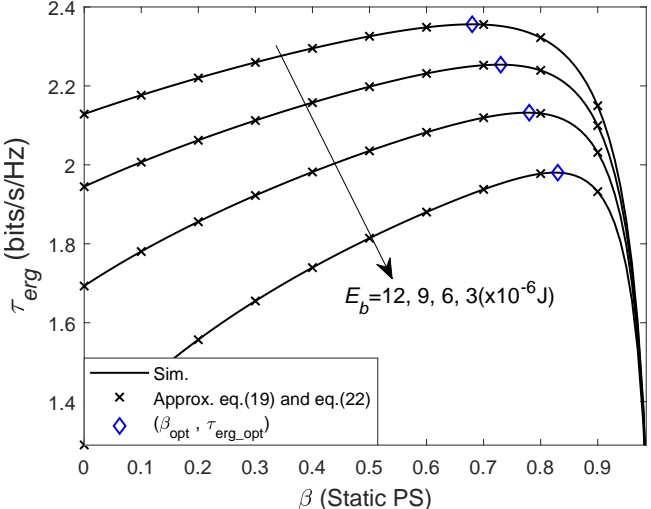

**Figure 6.** The ergodic capacity vs. $\beta$, where $\rho_s = 30$ dB.

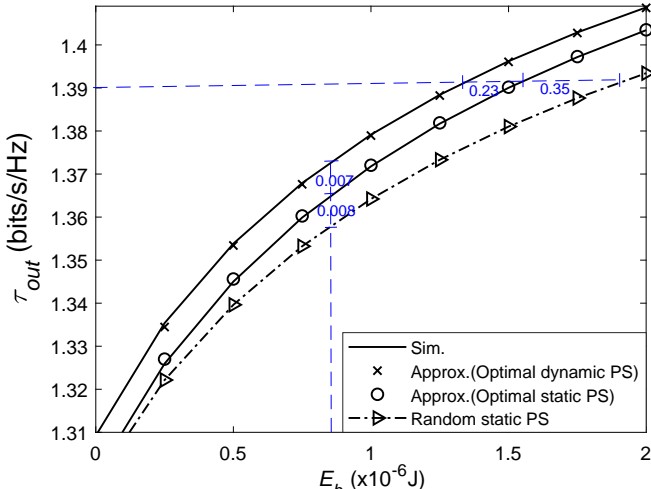

**Figure 7.** The outage capacity vs. $E_b$, where $\rho_s = 35$ dB and $Y = 3$ bits/s/Hz.

In Figure 8, the ergodic capacity against the assisted energy $E_b$ is investigated. The approximate results match the simulation results well, which verifies the accuracy of the derived ergodic capacity in Section 3.2 or Section 4.2. It can be observed that a higher ergodic capacity can be obtained under the dynamic PS by comparing with results achieved under the random or optimal static PS. The reason is that the dynamic PS ratio is adjusted based on the instantaneous CSI to maximize the overall SNR. Moreover, the random static PS achieves the lowest capacity among three schemes, which also suggests the importance of choosing an appropriated static PS ratio. Besides, it can also be seen that compared with the optimal static PS, a smaller assisted energy consumption is realized by the dynamic PS under the same ergodic capacity.

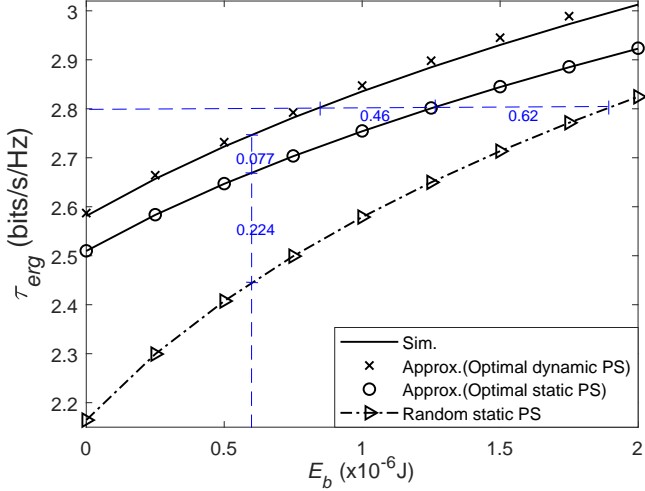

**Figure 8.** The ergodic capacity vs. $E_b$, where $\rho_s = 35$ dB.

Figure 9 illustrates the energy efficiency $EE_{out}$ versus the assisted energy $E_b$ for three PS schemes. The $EE_{out}$ is defined as $EE_{out} = \tau_{out}/(E_s + E_b)$, where $E_s = 0.5P_s$ and $E_b$ denote energy consumptions at the source and the relay, respectively. As can be observed from the figure, the $EE_{out}$ for three PS schemes reaches a peak one after another. After the peak, the $EE_{out}$ is getting lower. This indicates that although increasing $E_b$ increases the outage capacity, the $EE_{out}$ decreases when $E_b$ exceeds a threshold. It can also be observed that the $EE_{out}$ of the dynamic PS is higher than those under the random and optimal static PS. This is due to the fact that the dynamic PS achieves a higher outage capacity compared to the random or optimal static PS under the given total energy consumption.

Figure 10 represents the variations of the energy efficiency $EE_{erg}$ with the assisted energy $E_b$, where the $EE_{erg}$ is expressed as $EE_{erg} = \tau_{erg}/(E_s + E_b)$. We can see that the $EE_{erg}$ for three schemes firstly increases and then decreases. This suggests that the selection of appropriate $E_b$ is essential to balance spectral efficiency and energy efficiency. We can also see that the dynamic PS achieves the highest $EE_{erg}$ among these three schemes. This is because the dynamic PS achieves a higher ergodic capacity compared to the random or optimal static PS under the given total energy consumption.

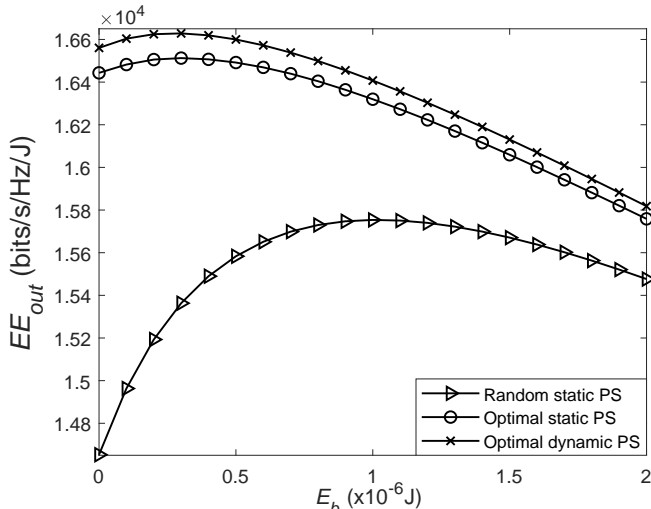

**Figure 9.** Energy efficiency for $\tau_{out}$ vs. $E_b$, where $\rho_s = 35$ dB and Y $= 3$ bits/s/Hz.

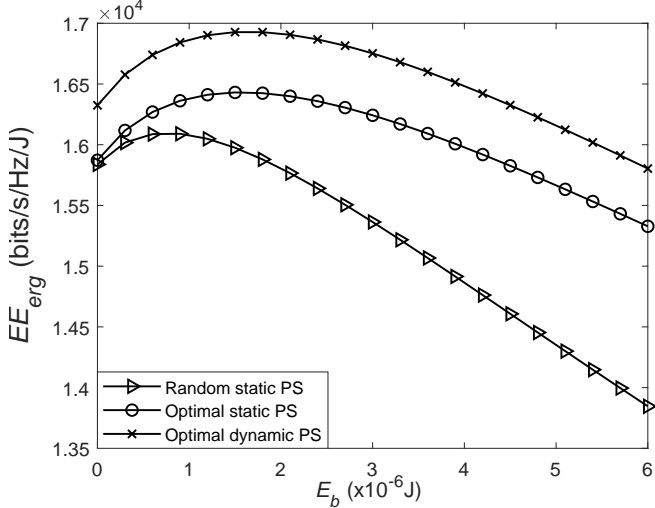

**Figure 10.** Energy efficiency for $\tau_{erg}$ vs. $E_b$, where $\rho_s = 35$ dB.

## 6. Conclusions

In this paper, by adopting a static/dynamic PS scheme at the battery-assisted SWIPT DF relay, the expressions for the outage and ergodic capacities were derived. We also compared the performance between the static and dynamic PS schemes. Some findings are summarized as follows. Firstly, the optimal static PS ratio to maximize the outage or ergodic capacity decreases as the assisted energy $E_b$ increases. Secondly, for a given $E_b$, the dynamic PS can achieve higher outage and ergodic capacities than the random or optimal static PS. Thirdly, compared with the optimal static PS, the dynamic PS consumes much less battery energy while achieving the same outage or ergodic capacity. Fourthly, we provide some insights on the selection of the assisted energy $E_b$ for the static and dynamic PS schemes in terms of energy efficiency.

Finally, we point out two possible future works. Firstly, it is interesting to consider the assumption that the harvested energy at the relay can be used to assist relaying transmission and recharge the battery. In this scenario, we can analyze the outage/ergodic performance in terms of statistical and instantaneous CSI, respectively. Secondly, the system model considered in this paper can be extended to the battery-assisted SWIPT enabled two-way relay system. Note that the system outage probability for the two-way relay system jointly takes the outage evens of both terminals into account, resulting in a high correlation between two links. Hence, quantifying the system outage performance for the two-way relay system is much more challenging than that for the dual-hop relay system.

**Author Contributions:** Formal analysis, Z.L. and Y.Y.; Funding acquisition, G.L.; Investigation, L.S.; Methodology, Y.Y.; Supervision, G.L.; Writing original draft, Z.L.; Writing review and editing, G.L., L.S. and Y.Y. All authors have read and agreed to the published version of the manuscript.

**Funding:** This research was funded by the Postgraduate Innovation Fund of Xi'an University of Posts and Telecommunications grant number CXJJLZ2019026.

**Conflicts of Interest:** The authors declare no conflict of interest.

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
