# Peer review of "On the Performance of Battery-Assisted PS-SWIPT Enabled DF Relaying"

_information, doi:10.3390/info11030165_

Round 1
Reviewer 1 Report
In this manuscript, expressions for the outage and ergodic capacities were derived for a static/dynamic PS scheme at the battery-assisted SWIPT DF relay. The topic is interesting. Here are some suggestions:
1. The introduction of the manuscript could be further modified. More related papers could be cited. Here are some examples:
Multiuser overhearing for cooperative two-way multiantenna relays, IEEE Transactions on Vehicular Technology, vol. 65, no. 5, May 2016.
Overhearing protocol design exploiting inter-cell interference in cooperative green networks, IEEE Transactions on Vehicular Technology, vol. 65, no. 1, Jan. 2016.
On outage minimization in RF energy harvesting relay assisted bidirectional communication, Wireless Networks, vol. 25, no. 7, Oct. 2019.
2. The system model in this manuscript is equavalent to that in the following paper in some special cases. It would be better to provide the simulation comparison for the outage probibility between approximation results in these two manuscipts.
Outage analysis for simultaneous wireless information and power transfer in dual-hop relaying networks, Wireless Networks, vol. 25, no. 2, Feb. 2019.
3. The approximation error could be further discussed.
4. Eq. (17)-(19) and (20)-(22) are misplaced.
Reviewer 2 Report
In this paper, the author claims that the performance of a battery-assisted power splitting-(PS-)SWIPT decode-and-forward (DF) relay system has not been studied. However, there are some works are introduced in the paper which are on outage probability of the system. Can you explain this ?Some similar works are not included in the paper, such as W. Jiao, G. Liu and H. Wu, "Queue Performance of Energy Harvesting Cognitive Radio Sensor Networks With Cooperative Spectrum Sharing," in IEEE Access, vol. 6, pp. 73548-73560, 2018.
Could you provide the final expressions of Eqs. (29) and (35) ?
Reviewer 3 Report
In this paper, the authors investigate the static and dynamic power splitting in the battery-assited SWIPT DF relaying system, and derived the expressions of the outage capacity and ergordic capacity. Simulation resuts verified the analytical results. In general, this paper is well organized and structured with rigorous mathematical derivations, and the results seems to be correct. The reviewer still have the following concerns.
It seems that the harvest energy is used for transmission only, however in battery assisted SWIPT, the harvested energy can also be used to recharge the battery to prolong the lifetime of the relay which is omitted in this paper. Please elaborate more on this point. In relay systems, it is a general case that there is also a direct link between the source and destination nodes. Of course the dual-hop relay system considered in this paper simplified the analysis, but what is the performance in the general case?Author Response
Please see the attachment.

Round 2
Reviewer 2 Report
My questions have been well addressed, and I not have further questions.
Reviewer 3 Report
The authors have addressed all my concerns, I do not have further comments.